# Coalitional Personalized Federated Learning: A Hedonic Game Perspective

## Abstract

This paper presents a novel coalitional personalized federated learning (CPFL) framework through a hedonic game model, enabling self-interested agents to form coalitions for learning. Departing from previous approaches limited to homogeneous priors over one-dimensional parameters, we address the more general case of heterogeneous priors. We characterize both socially optimal and stable coalition structures under two typical agent configurations: the atomic regime with equal sample size and non-atomic regime. We show that the optimization problems can be reduced to well-studied formulations, which are solvable by existing algorithms. Our key algorithmic contributions include BIdirectional-SCAN (BIS-CAN) and SPREAD, two algorithms for coalition structure formation satisfying both in-coalition stability and individual stability in each agent configuration. Furthermore, we discuss the optimality problem within high-dimensional parameter spaces, extending the one-dimensional theoretical results.

## 1 Introduction

Federated Learning (FL) (McMahan et al., 2017) has emerged in recent years as a prominent paradigm for collaborative machine learning. In this framework, each local data holder (or client) receives a global model from a central server, performs computations on its private dataset, and then sends individual model parameters back to the server for aggregation. This process iteratively updates the global model without requiring raw data to leave local devices. FL is particularly well-suited for scenarios where data is distributed and cannot be centralized due to privacy concerns or high communication costs, enabling global model training without direct access to user data.

In some real-world applications, local clients are not completely scheduled by a central server but instead act in a self-interested manner, aiming to seek individual models that perform well on their local data distributions. When clients have limited local data and cannot train a satisfactory model independently, they may join a FL process to benefit from a model trained on aggregated data from all participants. A good global model can typically be achieved when data is abundant and Independent and Identically Distributed (IID). However, in practice, clients often possess non-IID data with significant statistical heterogeneity due to variations in user demographics, behaviors, or local environments. This heterogeneity introduces major challenges in standard FL, including performance degradation (Pillutla et al., 2022; Xu et al., 2025), slow convergence (Li et al., 2020; Barona López & Borja Saltos, 2025), and unfairness (Li et al., 2023; Shen et al., 2025; Ray Chaudhury et al., 2024).

To address the challenges posed by data heterogeneity in FL, Personalized Federated Learning (PFL) has emerged as an extension where each client obtains a personalized model tailored to its unique data distribution while benefiting from collective knowledge. While PFL typically follows two main approaches: personalizing a global model for local clients and learning personalized models directly (Tan et al., 2022), this work focuses on the latter through a coalition-based approach, similar to the works of Yfantis et al. (2025) and Sattler et al. (2021). In our framework, clients are partitioned into coalitions based on data similarities, with each coalition training and sharing a specialized model among its members. This approach naturally captures the fundamental trade-off between **aggregation** (effective data volume) and **heterogeneity** (Non-IID degree).

To formally analyze this trade-off, we model the PFL problem as a hedonic game, where clients (players) form coalitions where the utility is solely based on the coalition membership. Through rigorous analysis of the federated mean estimation problem, we derive a utility function that naturally

separates into two components: a variance term representing the aggregation and a bias term for the heterogeneity. Within this game-theoretic framework, we examine both the **social optimality** of coalition structures, specifically those partitions that minimize total error across all agents, and the stability of partitions. Our stability analysis focuses on two key concepts: **in-coalition core stability**, which occurs when no subset of players in the same coalition can form a smaller coalition that at least a player can reduce its loss, and **individual stability**, which is achieved when no single player can improve its utility by joining another coalition without harming any existing agents of that coalition. We use the term 'in-coalition core stability' to distinguish it from the classical definition of 'core stability', which requires that no subset of the grand coalition exists where at least a player can achieve a lower loss.

**Contributions.** In this work, we focus on finding a socially optimal coalition structure and a stable coalition structure with one-dimensional parameters. Specifically, we examine two typical agent configurations: **atomic agents with equal sample size** and **non-atomic agents**. For atomic agents, we reduce the optimality problem to a regularized minimum sum-of-squares clustering (MSSC) problem, solvable via dynamic programming (Grønlund et al., 2018). For stability, the proposed algorithm BIdiretional-SCAN (BISCAN) constructs a coalition structure that is both in-coalition core stable and individually stable. In the non-atomic configuration, the optimality problem becomes a regularized optimal quantization problem, solvable iteratively via the Lloyd-MAX algorithm (Lloyd, 1982). Our proposed algorithm SPREAD ensures stability under the same two properties. With high-dimensional parameters, we show that no polynomial-time algorithm exists for the optimality problem in the atomic regime, while the LBG algorithm can be applied to the non-atomic regime.

## 1.1 RELATED WORK

**Hedonic Game.** Hedonic game has been introduced in Drèze & Greenberg (1980) and many different preferences are studied in the following works. Additively separable preference is a well-studied kind of hedonic preferences, where each agent assigns a value to every other agent and prefers coalitions with higher total utility. In particular, symmetric additively separable games always admit Nash-stable outcomes where no agent can benefit by moving to another coalition, and these can be found via local improvement dynamics using potential functions (Gairing & Savani, 2011). In contrast, core stability is NP-hard to verify even under this restricted model (Ballester, 2004). A recent work from Brandt et al. (2024) systematically analyzes the single agent deviations on forming stable coalitions.

Under top-responsive preferences, where players prefer coalitions that include more of their top-ranked peers, a core-stable partition always exists and can be computed using the Top Covering algorithm (Cechlárová & Hajduková, 2004). Further tractable subclasses include friend-enemy models (Aziz & Brandl, 2012), Boolean preference structures (Igarashi & Elkind, 2016), and anonymous preferences (Bogomolnaia & Jackson, 2002), each enabling efficient algorithms for certain stability concepts (e.g., contractual individual stability). To the best of our knowledge, however, the preference implied by our error function does not belong to any existing classes of preferences.

**Game Theoretical Analysis in Personalized Federated Learning.** Tan et al. (2022) provides a detailed review on PFL, summarizing two classical personalized federated learning methods: global model personalization and learning personalized models. The former focuses on improving the global model's performance on diverse data through methods like data augmentation and model regularization, while the latter aims to create tailored models for each client using approaches such as parameter decoupling and knowledge distillation.

However, most existing papers mentioned in (Tan et al., 2022) are experimental researches without much theoretical analysis. Blum et al. (2021) gives a theoretical analysis from the data contribution perspective. They model from some classical machine learning problems and analyze the existence of equilibria, the sample complexity of equilibria, and the price of stability and fairness. From the perspective of hedonic game by clustering agents into coalitions, which is a method of learning personalized models, Donahue & Kleinberg (2021b) model the agents within the federated system share the same prior distribution and hold different amount of data. Then Donahue & Kleinberg (2021b) analyze how to find the optimal coalition structure and a stable coalition structure with respect to certain stability concepts. However, these papers all focus on the homogeneous prior

model where all agents have the same prior distribution over their parameters, which is relaxed and extended in our model.

## 2 CPFL: PROBLEM FORMULATION AND PRELIMINARIES

In this section, we first introduce the basic setting of the personalized federated learning problem and introduce the concepts of partitions and coalitions. We then propose a game-theoretical framework to model the collaboration among all agents.

**Notation.** In this paper, for any $d, d_1, d_2 \in \mathbb{N}^+$ where $d_1 < d_2$ and $\mathbb{N}^+$ is the set of positive integers, we denote $[d]$ as the set $\{1, 2, \cdots, d\}$, and $[d_1 : d_2]$ as the set $\{d_1, d_1 + 1, \cdots, d_2\}$. A partition $\mathcal{C}$ of $[K]$ is a set of subsets $\mathcal{C} = \{C_1, ..., C_{|\mathcal{C}|}\}$ such that $\bigcup_{i=1}^{|\mathcal{C}|} C_i = [K]$ and $C_i \cap C_j = \emptyset$. We denote $\Pi_K$ as the set of all partitions over $[K]$. We use $\succeq_k$ to denote the complete and transitive preference relation of agent $k \in [K]$.

### 2.1 AGENTS AND COALITIONS

**Local agents.** We consider a personalized federated learning (PFL) problem with $K$ agents. Each agent $k \in [K]$ holds $n_k$ random sample $S_k = \{(\mathbf{x}_k^{(i)}, \mathbf{y}_k^{(i)})\}_{i \in [n_k]}$ drawn from her local distribution $\mathcal{D}_k$, where $(\mathbf{x}_k^{(i)}, \mathbf{y}_k^{(i)}) \in \mathbb{R}^D \times \mathbb{R}^L$ is the $i$-th data point. We assume that there exists a common loss function $\ell(\cdot, \cdot) : \mathbb{R}^L \times \mathbb{R}^L \to \mathbb{R}$ for all agents, and all agents share a common hypothesis space $\mathcal{H} = \{h(\cdot|\boldsymbol{\theta}) : \mathbb{R}^D \to \mathbb{R}^L, \boldsymbol{\theta} \in \mathbb{R}^d\}$ where the hypothesis $h$ is parameterized by $\boldsymbol{\theta}$. The goal of agent $k$ is to minimize the expected risk on her own distribution over parameter $\boldsymbol{\theta}$, where the expected risk is defined as $L_k(\boldsymbol{\theta}) = \mathbb{E}_{(\mathbf{x}_k, \mathbf{y}_k) \sim \mathcal{D}_k}[\ell(\mathbf{y}_k, h_{\boldsymbol{\theta}}(\mathbf{x}_k))]$. We define $\boldsymbol{\theta}_k = \arg\min_{\boldsymbol{\zeta}} L_k(\boldsymbol{\zeta})$ as the expected risk minimizer of agent $k$.

For agent $k$ and parameter $\hat{\boldsymbol{\theta}} \in \mathbb{R}^d$, we use the expected MSE $M_k(\hat{\boldsymbol{\theta}}) = \mathbb{E}[\|\hat{\boldsymbol{\theta}} - \boldsymbol{\theta}_k\|^2]$ to measure the performance of $\hat{\boldsymbol{\theta}}$, where the expectation is over all randomness of $\hat{\boldsymbol{\theta}}$. Following the work of Donahue & Kleinberg (2021b), we consider a Bayesian setting where each agent $k$ has a prior distribution $\mathcal{P}_k$ over the expected risk minimizer $\boldsymbol{\theta}_k$, where the distributions $\mathcal{P}_k$ are mutually independent across all agents. We denote $\mathcal{P} = \prod_{k \in [K]} \mathcal{P}_k$ as the joint prior distribution across all agents. Without knowing the expected risk minimizers, we focus on the prior expected MSE $E_k(\hat{\boldsymbol{\theta}}) = \mathbb{E}_{\mathcal{P}}[M_k(\hat{\boldsymbol{\theta}})]$ over the joint prior distribution to evaluate the performance of parameter $\hat{\boldsymbol{\theta}}$.

**Coalitions.** In this paper, a coalition $C$ is a non-empty subset of $[K]$. When receiving the parameters of all agents, the central server should decide a partition $\mathcal{C}$ and give a common model parameter to the agents in the same coalition. In this work, we assume that for any coalition $C \in \mathcal{C}$, the central server returns the weighted average model parameter $\hat{\boldsymbol{\theta}}_C = \sum_{k \in C} n_k \hat{\boldsymbol{\theta}}_k / N_C$ where $n_k$ is the sample size of agent $k$ and $N_C = \sum_{k \in C} n_k$. Hence, the prior expected MSE of an agent varies across different coalitions. To describe the property of a coalition and a coalition structure, with a slight notational ambiguity, we denote $E(C) = \sum_{k \in C} n_k E_k(\hat{\boldsymbol{\theta}}_C)$ as the coalition error over all agents given coalition $C$, and $E(\mathcal{C}) = \sum_{C \in \mathcal{C}} E(C)$ as the social error over all coalitions given partition $\mathcal{C}$.

### 2.2 EXACT ERROR ANALYSIS

From the theoretical side, as stated in Donahue & Kleinberg (2021b), we need a closed-form error function to analyze the optimality and the stability of a coalition structure. In the following section, we focus on a classic problem: mean estimation, which is considered in previous works (Donahue & Kleinberg, 2021a;b; 2023). We analyze the prior expected MSE of each agent in the Bayesian framework with given prior distributions. In addition, we also analyze the federated linear regression problem, and the results are given in the Appendix A.3 due to space limitations.

**Error Analysis in Mean Estimation.** In mean estimation problem, agent $k$ draws $n_k$ samples $\{\mathbf{y}_k^{(i)}\}_{i=1}^{n_k}$ from distribution $\mathcal{D}_k$ and wants to estimate the expectation vector $\boldsymbol{\mu}_k$. Following the assumptions in Donahue & Kleinberg (2021b) where the expectation vectors and the covariance

matrix are both assumed to be the same, we only assume that agents share a common covariance matrix denoted by $\boldsymbol{\Lambda}$, which is known to each agent. With local samples, the (prior) expected MSE for agent $k$ is $E_k(\hat{\boldsymbol{\mu}}_k) = M_k(\hat{\boldsymbol{\mu}}_k) = \text{tr}(\boldsymbol{\Lambda})/n_k$ when using the sample mean $\hat{\boldsymbol{\mu}}_k = \bar{\mathbf{y}}_k$. When collaborating in coalition $C$, the expected MSE for agent $k$ is:

**Lemma 2.1.** *In mean estimation problem, the expected MSE of agent $k$ in coalition $C$ is*

$$M_k(\hat{\boldsymbol{\mu}}_C) := \mathbb{E}[\|\hat{\boldsymbol{\mu}}_C - \boldsymbol{\mu}_k\|^2] = \underbrace{\|\boldsymbol{\mu}_C - \boldsymbol{\mu}_k\|^2}_{Bias\ Term} + \underbrace{\frac{\text{tr}(\boldsymbol{\Lambda})}{N_C}}_{Variance\ Term} . \tag{1}$$

In Lemma 2.1, we see that the MSE for agent $k$ when collaborating in coalition $C$ can be decomposed into the variance term and the bias term. With more agents entering the coalition, the variance term decreases since the denominator increases, but the bias term could increase if $\boldsymbol{\mu}_k$ is far from the coalition mean.

As stated before, we assume that agent $k$ has a prior distribution $\mathcal{P}_k$ over its true expectation vector $\boldsymbol{\mu}_k$. Specifically, in mean estimation problem, we assume that $\mathbb{E}_{\mathcal{P}_k}[\boldsymbol{\mu}_k] = \boldsymbol{\lambda}_k$ and $\text{cov}_{\mathcal{P}_k}[\boldsymbol{\mu}_k] = \boldsymbol{V}$ for all $k \in [K]$, where the covariance matrix $\boldsymbol{V}$ is common and known to each agent. The prior expected MSE for agent $k$ in coalition $C$ is:

**Lemma 2.2.** *In mean estimation problem, the prior expected MSE for agent $k$ in coalition $C$ is*

$$E_k(\hat{\boldsymbol{\mu}}_C) := \mathbb{E}_{\mathcal{P}}[M_k(\hat{\boldsymbol{\mu}}_C)] = \|\boldsymbol{\lambda}_C - \boldsymbol{\lambda}_k\|^2 + \frac{\sum_{i \in C, i \neq k} n_i^2 + (N_C - n_k)^2}{N_C^2} \text{tr}(\boldsymbol{V}) + \frac{\text{tr}(\boldsymbol{\Lambda})}{N_C}. \tag{2}$$

In (2), the first two terms on the right-hand side are the expectation of the bias term in (1) over the prior distribution, while the first term describes the distance of the prior coalition mean to the prior mean of agent $k$, and the second term is related to the sample size of each agent in coalition $C$.

## 2.3 FEDERATED HEDONIC GAME

Motivated by the mean estimation problem, we define a federated hedonic game where agents have preferences over the coalitions containing them. Formally, a hedonic game is a tuple $H = ([K], S, V, (n_k)_{k \in [K]}, (\mathbf{x}_k)_{k \in [K]}, E_k(\mathbf{x}_C)_{k \in C \subset [K]})$, where $[K]$ is the agent set, $S$ and $V$ are two constants. For any agent $k$, $\mathbf{x}_k$ is the prior expectation of her parameter, $n_k$ is the sample size, and the error function $E_k(\mathbf{x}_C)$ is defined as

$$E_k(\mathbf{x}_C) = \|\mathbf{x}_C - \mathbf{x}_k\|^2 + \frac{\sum_{i \in C, i \neq k} n_i^2 + (N_C - n_k)^2}{N_C^2} V + \frac{S}{N_C}, N_C := \sum_{i \in C} n_i. \tag{3}$$

Note that $C \succeq_k C'$ if and only if $E_k(\mathbf{x}_C) \leq E_k(\mathbf{x}_{C'})$. In hedonic game, a coalition structure $\mathcal{C}$ is a partition of $[K]$, and we use $\mathcal{C}(k)$ to denote the coalition that $k$ belongs to in partition $\mathcal{C}$.

For an agent $k$, if joining coalition $C$ decrease her MSE compared with local training, that is $E_k(\mathbf{x}_C) < E_k(\mathbf{x}_k) = S/n_k$, then she is willing to join and collaborate in coalition $C$. In a federated hedonic game, all agents send their parameters to the central server, then the server will decide and announce a coalition structure $\mathcal{C}$. Agents are allowed to deviate from the announced coalition they belong to. For example, each agent can make a request to join another coalition by reporting its own parameter, or some agents can collude to form a smaller coalition.

## 2.4 SOLUTION CONCEPTS

In this section, we list some ideal solution concepts in hedonic games. The first solution concept states the strongest property for a coalition structure, which is perfectness.

**Definition 2.3** (Perfectness). *A coalition structure $\mathcal{C}$ is perfect if for each agent $k$, $E_k(\mathbf{x}_{\mathcal{C}(k)}) = \min_{C \in [K]} E_k(\mathbf{x}_C)$.*

In a perfect coalition structure, all agents belong to their most-preferred coalitions among all coalitions containing her. Thus, a perfect coalition structure is always stable no matter what deviations. The next solution concept on a coalition structure is the social optimality.

**Definition 2.4** (Social Optimality). *A coalition structure $\mathcal{C}^{\text{opt}}$ is optimal in social welfare if it minimizes the social error across all agents: $\mathcal{C}^{\text{opt}} \in \arg\min_{\mathcal{C} \in \Pi_K} E(\mathcal{C})$.*

A socially optimal coalition structure is the states that a system designer wants to attain, but an optimal coalition structure may be unstable if some self-interest agents want to deviate. Next, we define two types of stability concepts.

**Definition 2.5** (In-Coalition Core Stability). *A coalition structure $\mathcal{C}$ is in-coalition core stable if there does not exist $C' \subset C \in \mathcal{C}$ such that for every agent $k \in C'$, $E_k(\mathbf{x}_{C'}) \leq E_k(\mathbf{x}_C)$ and for some agent $k' \in C'$, $E_{k'}(\mathbf{x}_{C'}) < E_{k'}(\mathbf{x}_C)$.*

An in-coalition core stable coalition structure describes the property that no subset of agents in the same coalition will form a smaller coalition to suffer lower error for the agents in this subset.

**Definition 2.6** (Individually Stability). *A coalition structure $\mathcal{C}$ is individually stable if for any $C \in \mathcal{C}$ and any $k \in C$, there does not exist $C' \in \mathcal{C}$, such that $E_k(\mathbf{x}_{C' \cup \{k\}}) < E_k(\mathbf{x}_C)$ and $E_j(\mathbf{x}_{C' \cup \{k\}}) \leq E_j(\mathbf{x}_{C'})$ for all $j \in C'$.*

In the following sections, we first focus on the algorithm design and analysis with **one-dimensional parameters** on two key problem: *(1)* how to find a social optimal coalition structure, *(2)* how to find a coalition structure that is both in-coalition core stable and individually stable. In the parts discussing about the one-dimensional parameters, we use light letter $x$ instead of the bold letter $\mathbf{x}$. Finally, we list some additional results with **high-dimensional parameters** in Section A.3.

## 3 OPTIMALITY AND STABILITY WITH ATOMIC AGENTS

In this section, we focus on the case where all agents hold the same amount of data. We will analyze how to find the optimal coalition structure and the coalition structure that is both in-coalition core stable and individually stable.

When all agents hold the same sample size $n$, we can simplify the error function 3 and decompose the variance term into the external variance and the internal variance:

$$E_k(x_C) = \underbrace{(x_C - x_k)^2}_{\text{Bias Term}} + \underbrace{\frac{T}{|C|}}_{\text{External Variance}} + \underbrace{V}_{\text{Internal Variance}} \text{, where } T = n^{-1}S - V. \tag{4}$$

In (4), $T$ describes the difference between the sampling variance and the parameter variance, and the internal variance is an inevitable error no matter which coalition an agent is in.

### 3.1 OPTIMALITY ANALYSIS

In this section, we discuss the optimal coalition structure which minimizes the total MSE across all agents. According to (3), the social error of a coalition structure $\mathcal{C}$ is

$$E(\mathcal{C}) = |\mathcal{C}|T + KV + \sum_{C \in \mathcal{C}} \sum_{k \in C} (x_k - x_C)^2. \tag{5}$$

First, we consider the trivial case where $T \leq 0$, that is, the sampling variance is smaller than the parameter variance. A perfectness conclusion is obtained immediately:

**Proposition 3.1.** *Conditioned on $T \leq 0$, the singleton coalition structure, where each agent forms a coalition individually, is perfect.*

When $T > 0$, the social optimality problem can be regarded as the one-dimensional Regularized Minimum Sum-of-Squares Clustering (Regularized-MSSC) problem. The MSSC problem is to find a coalition structure which minimizes the total squared Euclidean distances to each coalition mean, given that there are $l$ coalitions in total, where $l$ is an input parameter.

**Definition 3.2** (MSSC Problem, Grønlund et al. (2018)). *The MSSC problem is to solve*

$$V(l) = \min_{\mathcal{C} \in \Pi_K, |\mathcal{C}|=l} \sum_{C \in \mathcal{C}} \sum_{k \in C} \|\mathbf{x}_k - \mathbf{x}_C\|^2. \tag{6}$$

Hence, the social optimality problem regularizes on the number of coalitions of Problem 3.2. Actually, in Section 2.4 of Grønlund et al. (2018), there is a detailed analysis on solving the **one-dimensional** Regularized-MSSC problem. It is shown that solving the Regularized-MSSC problem of an input of size $K$ by Wilber algorithm (Wilber, 1988) takes $O(K)$ time.

### 3.2 STABILITY ANALYSIS AND BIDIRECTIONAL-SCAN (BISCAN)

In this section, we focus on two mentioned stability concepts: in-coalition core stability and individually stability. First, a natural question that arises is whether the optimal coalition structure has good stability properties? The following two propositions show the properties of the optimal coalition structure with respect to the two stability concepts.

**Proposition 3.3.** *For an optimal coalition structure $\mathcal{C}^{\mathrm{opt}}$, it holds that:*

*(a) $\mathcal{C}^{\mathrm{opt}}$ is in-coalition core stable.*

*(b) $\mathcal{C}^{\mathrm{opt}}$ can be not individually stable. For example, in a federated hedonic game instance $H = ([4], S = 13, V = 1, n = 1, (x_k) = (1, 2, 4, 6))$ with error function defined as (3), the optimal coalition structure is $\mathcal{C}^{\mathrm{opt}} = \{\{1, 2\}, \{3, 4\}\}$, but agent 3 can deviate and form $\{1, 2, 3\}$.*

Proposition 3.3 show that although the optimal coalition structure is in-coalition core stable, the individual stability cannot be guaranteed. Hence, we need to design new algorithms to find the coalition structure satisfying both the two stability properties. With the one-dimensional parameters, we sort the prior means of agents in ascending order and denote the agents as $x_1 \leq x_2 \leq ... \leq x_K$. Before introducing the algorithm, we first list two key properties with respect to the favorite coalition of the agent lying in the boundary of a coalition, which is quite useful in the algorithm design.

**Lemma 3.4.** *For agent $i$, there exists agents $r_i$ and $l_{r_i}$ satisfying $i \leq l_{r_i} \leq r_i$ such that:*

*(a) adding the agents on the right of agent $i$ one by one first decreases her error until agent $r_i$, then adding the agents on the right of $r_i$, if there exists such agent, increases her error. The coalition formed between agent $i$ and $r_i$ is the left-favorite coalition of agent $i$.*

*(b) $l_{r_i}$ is the left-most agent of the right-favorite coalition of $r_i$.*

Result (a) in Lemma 3.4 shows that for any agent $k$ lying on one of the boundary of a coalition, when adding agents on the other boundary, the error decreases first then increases; while result (b) in Lemma 3.4 states a "closed" property: for any coalition, if an agent on the boundary is left/right-favorite, then the coalition cannot allow more agents to be in since the error of at least one of the agents on the boundary will increase.

Now, we introduce our algorithm BISCAN, where the pseudo-code is presented in Algorithm 1. The general idea is the greedy strategy applied in both directions.

**Greedy search from left to right (line 2-6).** We start from the leftmost agent $l_1$ with parameter $x_1$. For each agent $l_i$ where $i \geq 1$, we iteratively find the left-favorite coalition $C_i$ whose right-most agent is $r_i$, then set $l_{i+1} = r_i + 1$. The left-favorite coalition for agent $l_i$ can be constructed by adding the agents on her right one by one from left to right, until the error of agent $l_i$ starts to increase, while the correctness is guaranteed by Lemma 3.4.

**Deviation from right to left (line 7-25).** When we reach the rightmost agent $K$, we need to check the individually stability from agent $r_{n-1}$. If the individually stability is guaranteed, we just return all the coalitions formed before; otherwise, we let the agents that have motivation to deviate move to the adjacent coalition, and iteratively check the motivation to deviate of the next agent. Whenever the next agent does not want to deviate, the algorithm terminates and returns all coalitions.

**Theorem 3.5.** *Algorithm 1 terminates in $O(K)$, and returns a coalition structure that is both in-coalition core stable and individually stable.*

## 4 OPTIMALITY AND STABILITY WITH NON-ATOMIC AGENTS

In this section, we consider the setting that there is a large population of agents. We focus on analyzing the limiting behavior where a large amount of agents form a coalition, each of which has

---

**Algorithm 1:** Bidirectional Scan

---

**Input:** Prior means $\{x_i\}_{i \in [K]}$ and $T$.

1 Set $\mathcal{G} = \emptyset$, $l_1 = r_1 = 1$, and $n = 0$.
2 **while** $r_n < K$ **do**
3      Update $n \leftarrow n + 1$.
4      Find the right-boundary agent $r_n \geq l_n$ of $l_n$'s left-favorite coalition.
5      Set $C_n = [l_n : r_n]$ and $l_{n+1} = r_n + 1$.
6 **end**
7 Find the left-boundary agent $l_{r_n} \leq r_n$ of $r_n$'s right-favorite coalition.
8 **if** $l_{r_n} \geq l_n$ **or** $r_{n-1}$ prefers $C_{n-1}$ to $C_n \cup \{r_{n-1}\}$ **then**
9      **return** $\mathcal{G} = \bigcup_{i=1}^{n} C_i$.
10 **else**
11      Update $C_n \leftarrow C_n \cup \{r_{n-1}\}$, set $k = n$ and $j = r_{k-1} - 1$.
12      **while** $k > 1$ **do**
13          **while** $j \geq l_{k-1}$ **and** $j$ prefers $C_k \cup \{j\}$ to $C_{k-1}$ **and** $r_k$ prefers $C_k \cup \{j\}$ to $C_k$ **do**
14              Update $C_k \leftarrow C_k \cup \{j\}$, $C_{k-1} \leftarrow C_{k-1} \backslash \{j\}$, and $j \leftarrow j - 1$.
15          **end**
16          **if** $j = l_{k-1} - 1$ **or** $r_{k-2}$ prefers $C_{k-2}$ to $C_{k-1} \cup \{r_{k-2}\}$ **or** $r_{k-1}$ prefers $C_{k-1}$ to $C_{k-1} \cup \{r_{k-2}\}$ **then**
17              **break**
18          **else**
19              Update $C_{k-1} \leftarrow C_{k-1} \cup \{r_{k-2}\}$ and $k \leftarrow k - 1$, set $j = r_{k-1} - 1$.
20          **end**
21      **end**
22 **end**
23 **return** $\mathcal{G} = \bigcup_{i=1}^{n} C_i$.

---

a very small amount of data. We will provide a new perspective of continuous agents different from the basic discrete setting in Section 2. But we see that the non-atomic model can also be directly derived from (3). Details can be found in Appendix A.4.

### 4.1 NON-ATOMIC AGENTS

**Mean estimation with non-atomic agents.** We assume that a large population of agents continuously distributed in the real line, in which every point $\mu$ can be regarded as an agent. To model the small amount setting, we consider the **sample density** instead of the sample size of each agent. Agent $\mu$ has sample density function $f(\mu)$, where $f(\mu)$ is positive, continuous, and light-tailed[1]. There are $K$ samples in total, that is, the integration of $f(\mu)$ over the real line equals $K$. For any $\mu$, there exists a corresponding random variable $X_\mu$ representing the coalition , where $\mathbb{E}[X_\mu] = e(\mu)$ and $\text{cov}[X_\mu, X_\eta] = S\delta(\mu - \eta)/f(\frac{\mu+\eta}{2})$, where $\delta(\cdot)$ is a Dirac delta function and $e(\mu)$ is unknown to agent $\mu$. This covariance term describes the property that any two agents are uncorrelated, and the variance of $X_\mu$ is $S\delta(0)/f(\mu)$, which is infinity since the data amount of an agent goes to zero. The assumption matches the result that the variance of the sample mean is equal to the sampling variance divided by the sample size. We denote that $F(a) = \int_{-\infty}^{a} f(\mu)d\mu$ and $G(a) = \int_{-\infty}^{a} \mu f(\mu)d\mu$, then we have $F'(a) = f(a)$ and $G'(a) = af(a)$. By the light-tailed assumption, every moment of the density distribution is finite for all $a \in \mathbb{R}$. We define the coalition centroid of a coalition $C$ as $X_C = \frac{\int_C X_\mu f(\mu)d\mu}{\int_C f(\mu)d\mu}$, which is the estimator for agents in coalition $C$. Similar to the standard setting in this paper, we also assume that agent $\mu$ has a prior knowledge on $e(\mu)$. That is, agent $\mu$ knows $\mathbb{E}[e(\mu)] = \mu$ and $\text{cov}[e(\mu), e(\eta)] = V\mathbb{I}[\mu = \eta]$.

**Lemma 4.1.** *The prior expected MSE for agent $\tau$ in coalition $C = [a, b]$ is*

$$E_\tau(X_{[a,b]}) = \frac{S}{F(b) - F(a)} + (\tau - H(a, b))^2 + V, \text{ where } H(a, b) = \frac{G(b) - G(a)}{F(b) - F(a)}. \quad (7)$$

---

[1]In this paper, we say a density $f(\mathbf{x})$ is light-tailed if $f(\mathbf{x}) \leq Ce^{-\alpha\|\mathbf{x}\|}$ for some constant $C$ and $\alpha$.

The proof of Lemma 4.1 can follow the proof with high-dimensional parameters, where details can be found in Appendix A.5.

## 4.2 OPTIMALITY ANALYSIS

In this section, we analyze how to find the social optimal coalition structure with non-atomic setting. With coalition structure $\mathcal{C} = \bigcup_{i=0}^{n}[\mu_i, \mu_{i+1}]$, where $\mu_0 = -\infty$ and $\mu_{|C|} = +\infty$, the social error $E(\mathcal{C}) := \sum_{C \in \mathcal{C}} E(C) = \sum_{C \in \mathcal{C}} \int_C f(\mu)E_\mu(C)$ is

$$E(\mathcal{C}) = (n+1)S + V + \int_{-\infty}^{\infty} \mu^2 f(\mu)d\mu - \sum_{i=0}^{n} H(\mu_i, \mu_{i+1})(G(\mu_{i+1}) - G(\mu_i)). \quad (8)$$

When the number of coalitions is fixed, minimizing $E(\mathcal{C})$ is equivalent to the one-dimensional Optimal Quantization problem, which maps continuous distributions into finite discrete data points.

**Definition 4.2** (Optimal Quantization, Pagès et al. (2004)). *For an $\mathbb{R}^d-$valued random vector $X$, the optimal quantization problem is to find the optimal measurable function $\phi(X)$ where $\phi$ takes at most $N$ values (quantizers) in $R^d$. Formally, the optimal quantization is to find out a measurable function $\phi^*$ such that*

$$\mathbb{E}[\|X - \phi^*(X)\|_2] = \inf\{\mathbb{E}[\|X - \phi(X)\|_2], \phi : \mathbb{R}^d \to \mathbb{R}^d, |\phi(\mathbb{R}^d)| \leq N\}.$$

The next lemma shows the optimality condition of the optimal coalition structure when the number of coalitions is fixed:

**Lemma 4.3.** *Fixed the number of coalitions $n+1$, the optimal coalition structure $\mathcal{C}_{n+1}^*$ with coalition boundaries $\{\mu_i\}_{i \in [n]}$ satisfies*

$$H(\mu_{i-1}, \mu_i) + H(\mu_i, \mu_{i+1}) = 2\mu_i. \quad (9)$$

*for all $i \in [n]$, where $\mu_0 = -\infty$ and $\mu_{n+1} = +\infty$.*

Lemma 4.3 states that the boundary must be the midpoint of the coalition centroids of two adjacent coalitions in the optimal coalition structure. Given the number of coalitions, we can use Lloyd-MAX (Algorithm 3), proposed in Lloyd (1982), to find an optimal coalition structure by iteratively updating the boundary and the coalition centroid. The existence of the optimal coalition structure and the convergence is also shown in Lloyd (1982). In Theorem 4.4, we show that the number of coalitions in the optimal coalition structure is upper bounded by some constant $n_0$, so we can iteratively add the number of coalitions to find the optimal coalition structure within finite rounds.

**Theorem 4.4.** *There exists $n_0 = n_0(S, \alpha)$, such that the number of coalitions $n$ in the optimal coalition structure is not larger than $n_0$.*

## 4.3 STABILITY ANALYSIS

In this section, we also attempt to find a coalition structure that is both in-coalition core stable and individually stable. The following lemma shows the optimal condition of an agent lying on the boundary of a coalition.

**Lemma 4.5.** *For agent $a \in \mathbb{R}$, the rightmost agent $b(a)$ in her left-favorite coalition satisfies*

$$\frac{S}{F(b(a)) - F(a)} = 2(H(a, b(a)) - a)(b(a) - H(a, b(a))). \quad (10)$$

*Similarly, for agent $b \in \mathbb{R}$, the leftmost agent $a(b)$ in her right-favorite coalition satisfies*

$$\frac{S}{F(b) - F(a(b))} = 2(H(a(b), b) - a(b))(b - H(a(b), b)). \quad (11)$$

Lemma 4.5 describes a crucial condition that if coalition $[a, b]$ is the left-favorite coalition of $a$, then it is also the right-favorite coalition of $b$ and vice versa. Based on this key property, we propose SPREAD, where the pseudo-code is in Algorithm 2.

SPREAD is quite simple: we start from any point $\mu$ in the real line, then we find the left-favorite and right-favorite coalitions in both directions, and iteratively form the coalitions by guaranteeing the optimality of the agents on the boundary.

---

**Algorithm 2:** SPREAD

---

**Input:** The sample density $f(\mu)$ and $S$.

1  Set $\mathcal{G} = \emptyset$
2  Randomly choose $\mu \in \mathbb{R}$, let $l_0 = u_0 = \mu$.
3  **while** *True* **do**
4  $\quad$ Find $l_{i+1} < l_i$ and $u_{i+1} > u_i$ such that

$$\frac{S}{F(l_{i+1}) - F(l_i)} = 2(H(l_i, l_{i+1}) - l_i)(l_{i+1} - H(l_i, l_{i+1})),$$

$\quad\quad$ and

$$\frac{S}{F(u_i) - F(u_{i+1})} = 2(H(u_{i+1}, u_i) - u_{i+1})(u_i - H(u_{i+1}, u_i)).$$

$\quad\quad \mathcal{G} \leftarrow \mathcal{G} \cup \{[l_i, l_{i+1}], [u_{i+1}, u_i]\}, i \leftarrow i + 1$
5  **end**

---

**Theorem 4.6.** *The coalition structure obtained from SPREAD is in-coalition core stable and individually stable.*

## 5   DISCUSSIONS IN OPTIMALITY WITH HIGH DIMENSIONAL PARAMETERS

**Atomic agents.**   First, we state the NP-hardness of the optimal coalition structure problem with high-dimensional parameters under atomic agents configuration. Mahajan et al. (2012) shows that MSSC problem is NP-hard when $d \geq 2$ for general $l$. Based on this result, we show that if $P \neq NP$, the optimality problem does not have polynomial algorithms when the parameters are high dimensional.

**Theorem 5.1.** *If $P \neq NP$, there do not exist polynomial algorithms to solve the optimal coalition structure problem when $d \geq 2$.*

**Non-atomic agents.**   In non-atomic setting with high-dimensional parameters, when the number of coalition is fixed, the problem is still the optimal quantization problem, and the optimal coalition structure must be a Voronoi region (Linde et al., 1980) where the boundary is the perpendicular bisector of line segment between two coalition means, which is similar to Lemma 4.3. The following theorem states the algorithm to find the optimal coalition structure.

**Theorem 5.2.** *When the number of coalitions is fixed, LBG algorithm Linde et al. (1980) converges to an optimal coalition structure. Moreover, when the density is light-tailed, the number of coalitions in the optimal coalitions is finite.*

More details can be found in the Appendix A.5.

## 6   CONCLUSION

In this paper, we mainly analyze the algorithm design on the formation of optimal coalition structure and stable coalition structure in two different settings: the atomic setting with equal amount of data and the non-atomic setting. We reduce the optimality problem into regularized MSSC problem and regularized quantization problem, and propose the BISCAN (Algorithm 1) and SPREAD (Algorithm 2) algorithms to find a stable coalition structure in each setting.

As for the future work, the stability analysis with high-dimensional parameters is still an open problem and deserves to be analyzed. Intuitively, we guess that the stable coalition structure may satisfy similar properties as the results with one-dimensional parameters such that the boundary agents should attain some optimality conditions. In addition, another interesting research direction is how to find a strict core stable coalition structure, which is a more common concept in hedonic game analysis.

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

## APPENDIX

### A.1 USAGE OF GENERATIVE AI

We use LLM for grammar checking.

## A.2 Pseudo-code of Lloyd-Max Algorithm

---

**Algorithm 3:** Lloyd-Max

---

**Input:** Probability density function $f(x)$, number of coalitions $n$, convergence threshold $\epsilon$
**Output:** Decision boundaries $\{b_i\}$ and reconstruction levels $\{y_i\}$

1   Initialize decision boundaries $\{\mu_0, \mu_1, \ldots, \mu_n\}$ with $\mu_0 = -\infty$, $\mu_n = \infty$;
2   Initialize reconstruction levels $\{y_1, y_2, \ldots, y_N\}$ arbitrarily;
3   **repeat**
4     **for** $i \leftarrow 1$ **to** $N-1$ **do**
5       $\mu_i \leftarrow \dfrac{y_i + y_{i+1}}{2}$ ;             // Update boundary
6     **end**
7     **for** $i \leftarrow 1$ **to** $N$ **do**
8       $y_i \leftarrow \dfrac{\int_{\mu_{i-1}}^{\mu_i} x f(x) dx}{\int_{\mu_{i-1}}^{\mu_i} f(x) dx}$ ;       // Centroid update
9     **end**
10   **until** $\max_i |y_i^{(t)} - y_i^{(t-1)}| < \epsilon$;

---

## A.3 Federated Linear Regression

In linear regression problem, agent $k$ draws $n_k$ samples $\{(X_k^i, y_k^i)\}_{i=1}^{n_k}$, where $X_k = [X_k^1; \ldots; X_k^{n_k}] \in \mathbb{R}^{n_k \times d}$ is the design matrix and $\mathbf{y}_k = (y_k^1, \ldots, y_k^{n_k}) \in \mathbb{R}^{n_k}$ is the response vector. We assume that $\mathbf{y}_k = X_k \boldsymbol{\beta}_k + \boldsymbol{\epsilon}_k$ and $\boldsymbol{\epsilon}_k | X_k \sim \mathcal{N}(\mathbf{0}, \sigma^2 I_{n_k})$ is a Gaussian noise and $X_k^i$ i.i.d. follows $\chi$ for all $i \in [n_k]$. With local samples, agent $k$ can use the Ordinary Least Squares (OLS) estimator $\hat{\boldsymbol{\beta}}_k = (X_k^\top X_k)^{-1} X_k^\top \mathbf{y}_k$ to estimate its parameter $\boldsymbol{\beta}_k$. Note that $\hat{\boldsymbol{\beta}}_k = \boldsymbol{\beta}_k + (X_k^\top X_k)^{-1} X_k^\top \boldsymbol{\epsilon}_k$, so $\hat{\boldsymbol{\beta}}_k | X_k$ follows $\mathcal{N}(\boldsymbol{\beta}_k, \sigma^2 (X_k^\top X_k)^{-1})$ and the MSE is $M_k(\{k\}) = \sigma^2 (X_k^\top X_k)^{-1}$. By Gauss–Markov theorem, the OLS estimator has the lowest sampling variance within the class of linear unbiased estimators. When collaborating in coalition $C$, the MSE for agent $k$ is:

**Lemma A.1.** *In linear regression problem, given the true regression coefficients, by choosing the weight $w_i = \mathrm{tr}(\mathbb{E}[(X_i^\top X_i)^{-1}])^{-1}$ in the federated average model, the MSE for agent $k$ in coalition $C$ is*

$$M_k(\hat{\boldsymbol{\beta}}_C) = \underbrace{\frac{\sigma^2}{W_C}}_{\text{Variance Term}} + \underbrace{\|\boldsymbol{\beta}_C - \boldsymbol{\beta}_k\|^2}_{\text{Bias Term}}. \tag{12}$$

*Proof.* Given the design matrices and the true regression coefficients, the expected MSE for agent $k$ in coalition $C$ is

$$\mathbb{E}[\|\hat{\boldsymbol{\beta}}_C - \boldsymbol{\beta}_k\|^2 | \{(\boldsymbol{\beta}_i, X_i)\}_{i \in C}]$$
$$= \mathbb{E}[\hat{\boldsymbol{\beta}}_C^\top \hat{\boldsymbol{\beta}}_C | \{(\boldsymbol{\beta}_i, X_i)\}_{i \in C}] + \boldsymbol{\beta}_k^\top \boldsymbol{\beta}_k - 2\boldsymbol{\beta}_k^\top \boldsymbol{\beta}_C$$
$$= \frac{1}{W_C^2} \sum_{i \in C} \sum_{j \in C} \mathbb{E}[w_i w_j \hat{\boldsymbol{\beta}}_i^\top \hat{\boldsymbol{\beta}}_j | (\boldsymbol{\beta}_i, X_i)] + \boldsymbol{\beta}_k^\top \boldsymbol{\beta}_k - 2\boldsymbol{\beta}_k^\top \boldsymbol{\beta}_C$$
$$= \frac{1}{W_C^2} \sum_{i \in C} \sum_{j \in C} w_i w_j \mathbb{E}[(\hat{\boldsymbol{\beta}}_i - \boldsymbol{\beta}_i)^\top (\hat{\boldsymbol{\beta}}_j - \boldsymbol{\beta}_j) | (\boldsymbol{\beta}_i, X_i)] + \|\boldsymbol{\beta}_C - \boldsymbol{\beta}_k\|^2$$

where the fourth equation holds since $\mathbb{E}[\hat{\boldsymbol{\beta}}_i|\boldsymbol{\beta}_i] = \boldsymbol{\beta}_i$ for all $i \in C$. By the equation $\hat{\boldsymbol{\beta}}_i = \boldsymbol{\beta}_i + (\boldsymbol{X}_i^\top \boldsymbol{X}_i)^{-1}\boldsymbol{X}_i^\top \boldsymbol{\epsilon}_i$ and the independence of $\hat{\boldsymbol{\beta}}_i$ and $\hat{\boldsymbol{\beta}}_j$, we have

$$
\mathbb{E}[\|\hat{\boldsymbol{\beta}}_C - \boldsymbol{\beta}_k\|^2 | \{(\boldsymbol{\beta}_i, \boldsymbol{X}_i)\}_{i \in C}]
$$
$$
= \frac{1}{W_C^2} \sum_{i \in C} w_i^2 \mathbb{E}[\boldsymbol{\epsilon}_i^\top \boldsymbol{X}_i (\boldsymbol{X}_i^\top \boldsymbol{X}_i)^{-2} \boldsymbol{X}_i^\top \boldsymbol{\epsilon}_i | \{(\boldsymbol{\beta}_i, \boldsymbol{X}_i)\}] + \|\boldsymbol{\beta}_C - \boldsymbol{\beta}_k\|^2
$$
$$
= \frac{1}{W_C^2} \sum_{i \in C} w_i^2 \mathbb{E}[\mathrm{tr}(\boldsymbol{X}_i (\boldsymbol{X}_i^\top \boldsymbol{X}_i)^{-2} \boldsymbol{X}_i^\top \boldsymbol{\epsilon}_i \boldsymbol{\epsilon}_i^\top) | (\boldsymbol{\beta}_i, \boldsymbol{X}_i)] + \|\boldsymbol{\beta}_C - \boldsymbol{\beta}_k\|^2
$$
$$
= \frac{1}{W_C^2} \sum_{i \in C} w_i^2 \mathrm{tr}(\boldsymbol{X}_i (\boldsymbol{X}_i^\top \boldsymbol{X}_i)^{-2} \boldsymbol{X}_i^\top) \mathbb{E}[\boldsymbol{\epsilon}_i \boldsymbol{\epsilon}_i^\top | \boldsymbol{\beta}_i] + \|\boldsymbol{\beta}_C - \boldsymbol{\beta}_k\|^2
$$
$$
= \frac{\sigma^2 \sum_{i \in C} w_i^2 \mathrm{tr}((\boldsymbol{X}_i^\top \boldsymbol{X}_i)^{-1})}{W_C^2} + \|\boldsymbol{\beta}_C - \boldsymbol{\beta}_k\|^2.
$$

and the second and the last step is because $\mathrm{tr}(\boldsymbol{A}\boldsymbol{B}) = \mathrm{tr}(\boldsymbol{B}\boldsymbol{A})$. Taking expectation over the randomness of the design matrix $\boldsymbol{X}_i$, we have

$$
M_k(C) = \frac{\sigma^2 \sum_{i \in C} w_i^2 \mathrm{tr}(\mathbb{E}[(\boldsymbol{X}_i^\top \boldsymbol{X}_i)^{-1}])}{W_C^2} + \|\boldsymbol{\beta}_C - \boldsymbol{\beta}_k\|^2
$$
$$
= \frac{\sigma^2}{W_C} + \|\boldsymbol{\beta}_C - \boldsymbol{\beta}_k\|^2
$$

by the choice of

$$
w_i = \mathrm{tr}(\mathbb{E}[(\boldsymbol{X}_i^\top \boldsymbol{X}_i)^{-1}])^{-1}.
$$

$\square$

Specifically, in linear regression, we assume that $\mathbb{E}[\boldsymbol{\beta}_k] = \boldsymbol{\lambda}_k^L$ and $\mathrm{cov}[\boldsymbol{\beta}_k] = \boldsymbol{V}^L$ for all $k \in [K]$; in mean estimation, we assume that $\mathbb{E}[\boldsymbol{\mu}_k] = \boldsymbol{\lambda}_k^L$ and $\mathrm{cov}[\boldsymbol{\mu}_k] = \boldsymbol{V}^L$ for all $k \in [K]$. We also assume that the true parameters are jointly independent across all agents.

**Lemma A.2.** *In linear regression problem, the prior expected MSE for agent $k$ in coalition $C$ is*

$$
E_k(\boldsymbol{\beta}_C) = \|\boldsymbol{\lambda}_C^L - \boldsymbol{\lambda}_k^L\|^2 + \frac{\sigma^2}{W_C} + \frac{\sum_{i \in C, i \neq k} w_i^2 + (\sum_{i \in C, i \neq k} w_i)^2}{W_C^2} \mathrm{tr}(\boldsymbol{V}^L).
$$

*Proof.* The prior expected MSE of the variance term is

$$
\mathbb{E}[\|\boldsymbol{\beta}_C - \boldsymbol{\beta}_k\|^2]
$$
$$
= \mathbb{E}[\|(\boldsymbol{\beta}_C - \boldsymbol{\lambda}_C^L) - (\boldsymbol{\beta}_k - \boldsymbol{\lambda}_k^L) + (\boldsymbol{\lambda}_C^L - \boldsymbol{\lambda}_k^L)\|^2]
$$
$$
= \mathbb{E}[\|\boldsymbol{\beta}_C - \boldsymbol{\lambda}_C^L\|^2] + \mathbb{E}[\|\boldsymbol{\beta}_k - \boldsymbol{\lambda}_k^L\|^2] + \|\boldsymbol{\lambda}_C^L - \boldsymbol{\lambda}_k^L\|^2
$$
$$
- 2\mathbb{E}[(\boldsymbol{\beta}_C - \boldsymbol{\lambda}_C^L)^\top (\boldsymbol{\beta}_k - \boldsymbol{\lambda}_k^L)]
$$

The last equation is because $\mathbb{E}[\boldsymbol{\beta}_k] = \boldsymbol{\lambda}_k^L$ and $\boldsymbol{\beta}_i$ and $\boldsymbol{\beta}_j$ are independent. By definition of $\boldsymbol{\beta}_C$ and $\boldsymbol{\lambda}_C^L$ and the independence again, we have that $\mathbb{E}[(\boldsymbol{\beta}_C - \boldsymbol{\lambda}_C^L)^\top (\boldsymbol{\beta}_k - \boldsymbol{\lambda}_k^L)] = w_k \mathbb{E}[\|\boldsymbol{\beta}_k - \boldsymbol{\lambda}_k^L\|^2]/W_C$,

then

$$\mathbb{E}[\|\boldsymbol{\beta}_C - \boldsymbol{\beta}_k\|^2]$$

$$= \frac{1}{W_C^2} \mathbb{E} \left[ \sum_{i \in C} \sum_{j \in C} w_i w_j (\boldsymbol{\beta}_i - \boldsymbol{\lambda}_i^L)^\top (\boldsymbol{\beta}_j - \boldsymbol{\lambda}_j^L) \right]$$

$$+ \left( 1 - \frac{2w_k}{W_C} \right) \mathbb{E}[\|\boldsymbol{\beta}_k - \boldsymbol{\lambda}_k^L\|^2] + \|\boldsymbol{\lambda}_C^L - \boldsymbol{\lambda}_k^L\|^2$$

$$= \frac{1}{W_C^2} \sum_{i \in C} w_i^2 \mathbb{E}[\|\boldsymbol{\beta}_i - \boldsymbol{\lambda}_i^L\|^2] + \left( 1 - \frac{2w_k}{W_C} \right) \operatorname{tr}(\boldsymbol{V}^L) + \|\boldsymbol{\lambda}_C^L - \boldsymbol{\lambda}_k^L\|^2$$

$$= \frac{\sum_{i \in C} w_i^2 \operatorname{tr}(\boldsymbol{V}^L)}{W_C^2} + \left( 1 - \frac{2w_k}{W_C} \right) \operatorname{tr}(\boldsymbol{V}^L) + \|\boldsymbol{\lambda}_C^L - \boldsymbol{\lambda}_k^L\|^2$$

$$= \frac{\sum_{i \in C, i \neq k} w_i^2 + (\sum_{i \in C, i \neq k} w_i)^2}{W_C^2} \operatorname{tr}(\boldsymbol{V}^L) + \|\boldsymbol{\lambda}_C^L - \boldsymbol{\lambda}_k^L\|^2.$$

Combining with lemma A.1 could complete the proof. $\qquad \square$

**Remark A.3.** *The choice of $w_i$ is natural since $\operatorname{tr}(\mathbb{E}[(\boldsymbol{X}_i^\top \boldsymbol{X}_i)^{-1}])^{-1}$ is proportional to the expected MSE of a local model, and the weighted average model minimizes the expected MSE when the local distributions are identical.*

**Remark A.4.** *The proof of Lemma 2.1 and Lemma 2.2 are quite similar and more simple, which are omitted.*

### A.4 ANOTHER PERSPECTIVE OF THE NON-ATOMIC AGENTS

We consider a set of agents are in the coalition $C$ with small amount of data. Specifically, let the sample size of agent $i$ in coalition $C$ is $n_{i,C}$. Then we assume that the sample size of each agent satisfies

$$\lim_{|C| \to \infty} \sup_{1 \leq i \leq |C|} n_{i,C} = 0, \quad \lim_{|C| \to \infty} \sum_{i=1}^{|C|} n_{i,|C|} = N > 0. \tag{13}$$

Based on Assumption 13 and (3), we have

$$\lim_{|C| \to \infty} E_k(\mathbf{x}_C) = \|\mathbf{x}_C - \mathbf{x}_k\|^2 + \frac{S}{N_C} + \lim_{|C| \to \infty} \frac{\sum_{i \in C, i \neq k} n_i^2 + (\sum_{i \in C, i \neq k} n_i)^2}{N_C^2} V$$

$$= \|\mathbf{x}_C - \mathbf{x}_k\|^2 + \frac{S}{N} + V.$$

### A.5 NON-ATOMIC SETTING WITH HIGH DIMENSIONAL PARAMETERS

**Mean estimation with continuous agents.** We consider a high-dimensional sample density $f(\boldsymbol{\mu})$, where $f(\boldsymbol{\mu})$ is positive, continuous, and light-tailed. For any $\boldsymbol{\mu}$, there exists a corresponding random variable $X_{\boldsymbol{\mu}}$, where $\mathbb{E}[X_{\boldsymbol{\mu}}] = e(\boldsymbol{\mu})$ and $\operatorname{cov}[X_{\boldsymbol{\mu}}, X_{\boldsymbol{\eta}}] = S\delta(\boldsymbol{\mu} - \boldsymbol{\eta})/f(\frac{\boldsymbol{\mu}+\boldsymbol{\eta}}{2})$, where $\delta(\cdot)$ is a Dirac delta function and $e(\boldsymbol{\mu})$ is unknown to agent $\boldsymbol{\mu}$. We denote that $F(\Delta) = \int_{\boldsymbol{\mu} \in \Delta} f(\boldsymbol{\mu}) d\boldsymbol{\mu}$ and $G(\Delta) = \int_{\boldsymbol{\mu} \in \Delta} \boldsymbol{\mu} f(\boldsymbol{\mu}) d\boldsymbol{\mu}$. Define the coalition centroid of a coalition $C$ as $X_C = \frac{\int_C X_{\boldsymbol{\mu}} f(\boldsymbol{\mu}) d\boldsymbol{\mu}}{\int_C f(\boldsymbol{\mu}) d\boldsymbol{\mu}}$.

The following lemma shows the expected MSE of agent $\boldsymbol{\mu}$ collaborating in coalition $C$.

**Lemma A.5.** *The MSE of agent $\tau$ in coalition $\Delta$ is*

$$M_{\boldsymbol{\tau}}(\Delta) = \frac{\operatorname{tr}(\mathbf{S})}{\int_C f(\boldsymbol{\mu}) d\boldsymbol{\mu}} + \left\| e(\boldsymbol{\tau}) - \frac{\int_C e(\boldsymbol{\mu}) f(\boldsymbol{\mu}) d\boldsymbol{\mu}}{\int_C f(\boldsymbol{\mu}) d\boldsymbol{\mu}} \right\|^2. \tag{14}$$

*Proof.* By definition,

$$\mathbb{E}\|X_C - e(\boldsymbol{\tau})\|^2 = \mathbb{E}\left\|\frac{\int_C X_{\boldsymbol{\mu}} f(\boldsymbol{\mu})d\boldsymbol{\mu}}{\int_C f(\boldsymbol{\mu})d\boldsymbol{\mu}} - e(\boldsymbol{\tau})\right\|^2$$

$$= \frac{1}{(\int_C f(\boldsymbol{\mu})d\boldsymbol{\mu})^2}\mathbb{E}\left\|\int_C (X_{\boldsymbol{\mu}} - e(\boldsymbol{\tau}))f(\boldsymbol{\mu})d\boldsymbol{\mu}\right\|^2$$

$$= \frac{1}{(\int_C f(\boldsymbol{\mu})d\boldsymbol{\mu})^2}\int_C \int_C \mathbb{E}[(X_{\boldsymbol{\mu}} - e(\boldsymbol{\tau}))^\top (X_{\boldsymbol{\eta}} - e(\boldsymbol{\tau}))]f(\boldsymbol{\mu})f(\boldsymbol{\eta})d\boldsymbol{\mu}d\boldsymbol{\eta}$$

$$= \frac{1}{(\int_C f(\boldsymbol{\mu})d\boldsymbol{\mu})^2}\int_C \int_C (\mathbb{E}[(X_{\boldsymbol{\mu}} - e(\boldsymbol{\mu}))^\top (X_{\boldsymbol{\eta}} - e(\boldsymbol{\eta}))] + (e(\boldsymbol{\mu}) - e(\boldsymbol{\tau}))(e(\boldsymbol{\eta}) - e(\boldsymbol{\tau})))f(\boldsymbol{\mu})f(\boldsymbol{\eta})d\boldsymbol{\mu}d\boldsymbol{\eta}$$

$$= \frac{1}{(\int_C f(\boldsymbol{\mu})d\boldsymbol{\mu})^2}\left[\int_C \int_C \frac{\mathrm{tr}(\mathbf{S})\delta(\boldsymbol{\mu} - \boldsymbol{\eta})}{f(\frac{\boldsymbol{\mu}+\boldsymbol{\eta}}{2})}f(\boldsymbol{\mu})f(\boldsymbol{\eta})d\boldsymbol{\mu}d\boldsymbol{\eta} + \left\|\int_C (e(\boldsymbol{\mu}) - e(\boldsymbol{\tau}))f(\boldsymbol{\mu})d\boldsymbol{\mu}\right\|^2\right]$$

$$= \frac{\mathrm{tr}(\mathbf{S})}{\int_C f(\boldsymbol{\mu})d\boldsymbol{\mu}} + \left\|e(\boldsymbol{\tau}) - \frac{\int_C e(\boldsymbol{\mu})f(\boldsymbol{\mu})d\boldsymbol{\mu}}{\int_C f(\boldsymbol{\mu})d\boldsymbol{\mu}}\right\|^2,$$

which completes the proof. $\square$

The MSE has similar formula to that in Lemma 2.1, the variance term is inversely proportional to the total sample size in coalition $[a, b]$, and the bias term describes the distance between agent $\mu$ and the coalition centroid.

**Prior distribution.** Similar to the standard setting in this paper, we also assume that agent $\boldsymbol{\mu}$ has a prior knowledge on $e(\boldsymbol{\mu})$. That is, agent $\mu$ knows $\mathbb{E}[e(\boldsymbol{\mu})] = \boldsymbol{\mu}$ and $\mathrm{cov}[e(\boldsymbol{\mu}), e(\boldsymbol{\eta})] = V\mathbb{I}[\boldsymbol{\mu} = \boldsymbol{\eta}]$.

Define the following function

$$H(C) = \frac{\int_C \boldsymbol{\mu} f(\boldsymbol{\mu})d\boldsymbol{\mu}}{\int_C f(\boldsymbol{\mu})d\boldsymbol{\mu}}. \tag{15}$$

**Lemma A.6.** *The prior expected MSE for agent $\tau$ in coalition $C$ is*

$$E_{\boldsymbol{\tau}}(C) = \mathrm{tr}(\mathbf{V}) + \frac{\mathrm{tr}(\mathbf{S})}{\int_C f(\boldsymbol{\mu})d\boldsymbol{\mu}} + \|e(\boldsymbol{\tau}) - H(C)\|^2. \tag{16}$$

*Proof.* Actually, we only need to calculate

$$\mathbb{E}\left[\left\|\int_C (e(\boldsymbol{\tau}) - e(\boldsymbol{\mu}))f(\boldsymbol{\mu})d\boldsymbol{\mu}\right\|^2\right]$$

$$= \int_C \int_C \mathbb{E}[(e(\boldsymbol{\tau}) - e(\boldsymbol{\mu}))^\top (e(\boldsymbol{\tau}) - e(\boldsymbol{\eta}))]f(\boldsymbol{\mu})f(\boldsymbol{\eta})d\boldsymbol{\mu}d\boldsymbol{\eta}$$

$$= \int_C \int_C ((\boldsymbol{\tau} - \boldsymbol{\mu})^\top (\boldsymbol{\tau} - \boldsymbol{\eta}) + \mathrm{tr}(\mathbf{V}))f(\boldsymbol{\mu})f(\boldsymbol{\eta})d\boldsymbol{\mu}d\boldsymbol{\eta}$$

$$= \mathrm{tr}(\mathbf{V}) + \|\boldsymbol{\tau} - H(C)\|^2,$$

which completes the proof. $\square$

The total prior expected MSE of a coalition $[a, b]$ is

$$E(C) = \int_a^b f(\boldsymbol{\mu})E_{\boldsymbol{\mu}}(C) = \mathrm{tr}(\mathbf{S}) + F(C)\mathrm{tr}(\mathbf{V}) + \left\|e(\boldsymbol{\tau}) - \frac{\int_C e(\boldsymbol{\mu})f(\boldsymbol{\mu})d\boldsymbol{\mu}}{\int_C f(\boldsymbol{\mu})d\boldsymbol{\mu}}\right\|^2. \tag{17}$$

The Linde–Buzo–Gray (LBG) algorithm (Algorithm 4) proposed by Linde et al. (1980) is shown below, which can be used to solve the vector quantization problem when the number of coalitions is given.

---

**Algorithm 4:** Linde–Buzo–Gray (LBG) Algorithm for Continuous Density Input

---

**Input:** Target codebook size $K$; Initial codebook $\mathcal{C}^{(0)} = \{c_1^{(0)}, \ldots, c_K^{(0)}\}$;
    Density function $f(x)$ on $\mathbb{R}^d$; Convergence threshold $\epsilon$
**Output:** Optimal codebook $\mathcal{C} = \{c_1, \ldots, c_K\}$ minimizing expected MSE

1 $t \leftarrow 0$; Initialize distortion $\mathcal{E}^{(-1)} \leftarrow \infty$;
2 **repeat**
  // Step 1: Nearest-Neighbor Partition (Voronoi regions)
3  Define regions $\mathcal{R}_i^{(t)} = \left\{ x \in \mathbb{R}^d : \|x - c_i^{(t)}\|^2 \le \|x - c_j^{(t)}\|^2, \forall j \right\}$
  // Step 2: Centroid Update
4  **for** $i \leftarrow 1$ **to** $K$ **do**
5

$$c_i^{(t+1)} = \frac{\int_{\mathcal{R}_i^{(t)}} x \, f(x) \, dx}{\int_{\mathcal{R}_i^{(t)}} f(x) \, dx} \quad \text{(centroid of region under } f)$$

6  **end**
  // Step 3: Compute Expected Distortion
7

$$\mathcal{E}^{(t)} = \sum_{i=1}^{K} \int_{\mathcal{R}_i^{(t)}} \|x - c_i^{(t+1)}\|^2 f(x) \, dx$$

8  $t \leftarrow t + 1$
9 **until** $|\mathcal{E}^{(t)} - \mathcal{E}^{(t-1)}| < \epsilon$;
10 **return** $\mathcal{C} = \{c_1^{(t)}, \ldots, c_K^{(t)}\}$

---

In addition, with nearly the same idea as the proof of Theorem 4.4, we can also show that the number of coalitions in the optimal coalition structure is finite, so the details are omitted.

### A.6 OMITTED PROOF

**Proposition 3.1.** *Conditioned on $T \le 0$, the singleton coalition structure, where each agent forms a coalition individually, is perfect.*

*Proof.* Since $T \le 0$, the external variance is minimized when $|C| = 1$ and the bias term is minimized when $\theta_C = \theta_k$. Hence, we see that the singleton coalition structure satisfies the two above optimality conditions, which completes our proof.

$\square$

**Proposition 3.3.** *For an optimal coalition structure $\mathcal{C}^{\text{opt}}$, it holds that:*

*(a) $\mathcal{C}^{\text{opt}}$ is in-coalition core stable.*

*(b) $\mathcal{C}^{\text{opt}}$ can be not individually stable. For example, in a federated hedonic game instance $H = ([4], S = 13, V = 1, n = 1, (x_k) = (1, 2, 4, 6))$ with error function defined as (3), the optimal coalition structure is $\mathcal{C}^{\text{opt}} = \{\{1, 2\}, \{3, 4\}\}$, but agent 3 can deviate and form $\{1, 2, 3\}$.*

*Proof.* We only proof (a) by contradiction. (b) can be easily verified by definition.

Assume that $C' \subset C$ has motivation to leave $C$ and form a new coalition, then for some agent $i \in C'$, we have

$$\frac{T}{|C'|} + \|\mathbf{x}_i - \mathbf{x}_{C'}\|^2 < \frac{T}{|C|} + \|\mathbf{x}_i - \mathbf{x}_C\|^2.$$

By summing the inequalities on all agents in $C'$, we have

$$T + \sum_{i \in C'} \|\mathbf{x}_i\|^2 - |C'|\|\mathbf{x}_{C'}\|^2 < \frac{|C|'}{|C|}T + \sum_{i \in C'} \|\mathbf{x}_i\|^2 + |C'|(\|\mathbf{x}_C\|^2 - 2\mathbf{x}_C^\top \mathbf{x}_{C'}),$$

that is,

$$\|\mathbf{x}_C - \mathbf{x}_{C'}\|^2 > \left(\frac{1}{|C'|} - \frac{1}{|C|}\right) T.$$

Since $C$ is in the optimal coalition structure, then $E(C) \leq E(C') + E(C \backslash C')$, i.e.,

$$T + \sum_{i \in C} \|\mathbf{x}_i - \mathbf{x}_C\|^2 \leq 2T + \sum_{i \in C'} \|\mathbf{x}_i - \mathbf{x}_{C'}\|^2 + \sum_{i \in C \backslash C'} \|\mathbf{x}_i - \mathbf{x}_{C \backslash C'}\|^2$$

$$\sum_{i \in C} \|\mathbf{x}_i\|^2 - |C|\|\mathbf{x}_C\|^2 \leq T + \sum_{i \in C} \|\mathbf{x}_i\|^2 - |C'|\|\mathbf{x}_{C'}\|^2 - (|C| - |C'|)\|\mathbf{x}_{C \backslash C'}\|^2$$

$$-|C|\|\mathbf{x}_C\|^2 \leq T - |C'|\|\mathbf{x}_{C'}\|^2 - (|C| - |C'|)\|\mathbf{x}_{C \backslash C'}\|^2.$$

Note that

$$\mathbf{x}_{C \backslash C'} = \frac{|C|\mathbf{x}_C - |C'|\mathbf{x}_{C'}}{|C| - |C'|},$$

we have

$$T \geq |C'|\|\mathbf{x}_{C'}\|^2 + \frac{|C|^2\|\mathbf{x}_C\|^2 + |C'|^2\|\mathbf{x}_{C'}\|^2 - 2|C||C'|\mathbf{x}_C^\top \mathbf{x}_{C'}}{|C| - |C'|} - |C|\|\mathbf{x}_C\|^2$$

$$= \frac{|C||C'|\|\mathbf{x}_C - \mathbf{x}_{C'}\|^2}{|C| - |C'|}.$$

Thus,

$$\|\mathbf{x}_C - \mathbf{x}_{C'}\|^2 \leq \left(\frac{1}{|C'|} - \frac{1}{|C|}\right) T,$$

which leads to contradiction. $\qquad\square$

**Lemma 3.4.** *For agent $i$, there exists agents $r_i$ and $l_{r_i}$ satisfying $i \leq l_{r_i} \leq r_i$ such that:*

*(a) adding the agents on the right of agent $i$ one by one first decreases her error until agent $r_i$, then adding the agents on the right of $r_i$, if there exists such agent, increases her error. The coalition formed between agent $i$ and $r_i$ is the left-favorite coalition of agent $i$.*

*(b) $l_{r_i}$ is the left-most agent of the right-favorite coalition of $r_i$.*

*Proof.* **Proof of (a).** We focus on a sequence $\{x_1, \cdots, x_n\}$ such that $x_1 \leq \cdots \leq x_n$, define $f_m = \frac{T}{m} - 2(\bar{x}_{[m]} - x_1)(x_m - \bar{x}_{[m]})$ for all $m \in [n]$, and we will focus on the error analysis of agent 1 when it lies in the left-most of a coalition.

A key term is the error difference when an agent is added into the coalition:

$$E_1(x_{[m]}) - E_1(x_{[m-1]}) = \left(\frac{T}{m} + (x_m - \bar{x}_m)^2\right) - \left(\frac{T}{m-1} + \left(x_m - \frac{m\bar{x}_m - x_1}{m-1}\right)^2\right)$$

$$= -\frac{1}{m-1}\left[\frac{T}{m} - 2(x_m - \bar{x}_m)(\bar{x}_m - x_1) + \frac{(\bar{x}_m - x_1)^2}{m-1}\right]$$

$$= -\frac{1}{m-1}\left[f_m + \frac{(\bar{x}_m - x_1)^2}{m-1}\right]$$

so when $f_m + \frac{(\bar{x}_m - x_1)^2}{m-1} > 0$, adding agent $m$ into the coalition decreases the error of agent 1.

**Lemma A.7.** *If $f_s \leq 0$ for some $s \in [2 : n]$, then there exists $l \in [2, s]$ such that $f_1 \geq \cdots \geq f_{l-1} \geq 0$, and $f_l \leq 0$.*

*Proof.* If $f_m \geq 0$, we have that

$$f_m - f_{m+1} = [\frac{T}{m} - 2(x_m - \bar{x}_m)(\bar{x}_m - x_1)] - [\frac{T}{m+1} - 2(x_{m+1} - \bar{x}_{m+1})(\bar{x}_{m+1} - x_1)]$$

$$= \frac{T}{m(m+1)} - 2(x_m - \bar{x}_m)(\bar{x}_m - x_1) + \frac{2m}{(m+1)^2}(x_{m+1} - \bar{x}_m)(m\bar{x}_m + x_{m+1} - (m+1)x_1)$$

$$\geq \frac{T}{m(m+1)} - 2(x_m - \bar{x}_m)(\bar{x}_m - x_1) + \frac{2m}{(m+1)^2}(x_m - \bar{x}_m)(m\bar{x}_m + x_m - (m+1)x_1)$$

$$= \frac{1}{m+1}\left[f_m + \frac{2m}{m+1}(x_m - \bar{x}_m)^2\right] \geq 0,$$

so we have $f_m \geq f_{m+1}$. Hence, there exists $l \in [2, n]$, such that $f_1 \geq f_2 \geq \cdots \geq f_{l-1}$, and $f_l < 0$. $\qquad\square$

**Lemma A.8.** *If $f_l \leq 0$, it holds that $f_{l+1} + \frac{(x_{l+1} - \bar{x}_{l+1})^2}{l} \leq 0$.*

*Proof.*

$$f_{l+1} + \frac{(x_{l+1} - \bar{x}_{l+1})^2}{l}$$

$$= \frac{T}{l+1} - 2(x_{l+1} - \bar{x}_{l+1})(\bar{x}_{l+1} - x_1) + \frac{(x_{l+1} - \bar{x}_{l+1})^2}{l}$$

$$= \frac{l}{l+1}\left[\frac{T}{l} - \frac{2}{l+1}(x_{l+1} - \bar{x}_l)(l\bar{x}_l + x_{l+1} - (l+1)x_1) + \frac{(x_{l+1} - \bar{x}_l)^2}{l+1}\right]$$

$$\leq \frac{l}{l+1}\left[2(x_l - \bar{x}_l)(\bar{x}_l - x_1) - \frac{2}{l+1}(x_{l+1} - \bar{x}_l)(l\bar{x}_l + x_{l+1} - (l+1)x_1) + \frac{(x_{l+1} - \bar{x}_l)^2}{l+1}\right]$$

$$= \frac{l}{(l+1)^2}\left[2(l+1)(x_l - \bar{x}_l)(\bar{x}_l - x_1) - 2l(x_{l+1} - \bar{x}_l)(\bar{x}_l - x_1) + (\bar{x}_l - x_1)^2 - (x_{l+1} - x_1)^2\right]$$

$$\leq \frac{l}{(l+1)^2}\left[2(l+1)(x_l - \bar{x}_l)(\bar{x}_l - x_1) - 2l(x_l - \bar{x}_l)(\bar{x}_l - x_1) + (\bar{x}_l - x_1)^2 - (x_l - x_1)^2\right]$$

$$= -\frac{l}{(l+1)^2}(x_l - \bar{x}_l)^2 \leq 0.$$

$\qquad\square$

**Lemma A.9.** *If $f_l + \frac{(x_l - \bar{x}_l)^2}{l-1} \leq 0$, then $f_{l+s} + \frac{(x_{l+s} - \bar{x}_{l+s})^2}{l+s-1} \leq 0$ for any $s \in \mathbb{N}^+$.*

*Proof.* Based on lemma A.8, since $f_l \leq f_l + \frac{(x_l - \bar{x}_l)^2}{l-1} \leq 0$, we have $f_{l+1} + \frac{(x_{l+1} - \bar{x}_{l+1})^2}{l} \leq 0$. By induction, we conclude that $f_{l+s} + \frac{(x_{l+s} - \bar{x}_{l+s})^2}{l+s-1} \leq 0$ for any $s \in \mathbb{N}^+$. $\qquad\square$

Lemma A.7, A.8, and A.9 claims the following property:

**Corollary .9.1.** *The right-most agent in the left-favorite coalition of agent $1$ is the agent $l$ satisfying $f(l) \leq 0$ and $f_l + \frac{(x_l - \bar{x}_{1:l})^2}{l-1} \geq 0$.*

*Proof.* By lemma A.7 and lemma 3.4, for any agent $i$ such that $f_i > 0$, forming a continuous coalition between agent 1 and agent $i$ will decrease the error of agent 1. For agent $l$ such that $f_l \leq 0$ and $f_l + \frac{(x_l - \bar{x}_{1:l})^2}{l-1} \geq 0$, agent $l$ should also be added into the coalition. But since $f_{l+1} \leq 0$ and $f_{l+1} + \frac{(x_{l+1} - \bar{x}_{1:l+1})^2}{l} \leq 0$, all agents that are larger than $l$ should not be added into the coalition, which completes the proof. $\qquad\square$

Actually, Corollary .9.1 completes the proof.

**Proof of (b)**. By definition and Corollary .9.1, $r_i$ satisfies that

$$\frac{T}{r_i - i + 1} - 2(\bar{\theta}_{[i:r_i]} - \theta_i)(\theta_{r_i} - \bar{\theta}_{[i:r_i]}) \leq 0.$$

Hence, by Lemma A.8, we obtain

$$\frac{T}{r_i - i + 2} - 2(\bar{\theta}_{(i-1):r_i} - \theta_{i-1})(\theta_{r_i} - \bar{\theta}_{(i-1):r_i}) \leq 0,$$

so $l_{r_i} \geq i$. $\square$

**Theorem 3.5.** *Algorithm 1 terminates in $O(K)$, and returns a coalition structure that is both in-coalition core stable and individually stable.*

*Proof.* The time complexity can be directly obtained sine the algorithm only compute on each agent at most twice.

The weak core stability always holds since all boundary agents satisfy the condition in Lemma .9.1. Next, we only check the individually stability.

By Lemma 3.4, in the first phase of Algorithm 1, without taking agents in coalition $C_n$ into consideration, all other agents cannot deviate. For example, if $r_i$ has motivation to move into $C_{i+1}$, but it will increase the error of $r_{i+1}$, so this deviation will not be acceptable.

Now we focus on coalition $C_n$. Whenever the adjacent agent of coalition $C_n$ has motivation to move into $C_n$, while agent $K$ accepts, a legal deviation will happen. Note that the possible deviation will at most reach agent $l_{n-1}$ by the construction in the first phase. If agent $l_{n-1}$ deviates to coalition $C_n$, we note that there is no longer any possible deviation. Otherwise, some agents will be left in $C_{n-1}$.

Iteratively, whenever the left-adjacent agent of coalition $C_{n-1}$ has motivation to move into $C_{n-1}$, while the right-most agent in coalition $C_{n-1}$ accepts, a legal deviation will happen.

The iteration will end after at most $n-1$ rounds when it reach coalition $C_1$, then the algorithm ends and there does not exist any legal deviation. $\square$

**Lemma 4.3.** *Fixed the number of coalitions $n+1$, the optimal coalition structure $\mathcal{C}^*_{n+1}$ with coalition boundaries $\{\mu_i\}_{i \in [n]}$ satisfies*

$$H(\mu_{i-1}, \mu_i) + H(\mu_i, \mu_{i+1}) = 2\mu_i. \tag{9}$$

*for all $i \in [n]$, where $\mu_0 = -\infty$ and $\mu_{n+1} = +\infty$.*

*Proof.* Based on (8), we calculate the first order condition on $\mu_1, \cdots, \mu_n$,

$$\frac{\partial E(\mathcal{C})}{\partial \mu_i} = -\frac{-2\mu_i f(\mu_i)(G(\mu_{i+1}) - G(\mu_i))(F(\mu_{i+1}) - F(\mu_i)) + (G(\mu_{i+1}) - G(\mu_i))^2 f(\mu_i)}{(F(\mu_{i+1}) - F(\mu_i))^2}$$

$$- \frac{2\mu_i f(\mu_i)(G(\mu_i) - G(\mu_{i-1}))(F(\mu_i) - F(\mu_{i-1})) - (G(\mu_i) - G(\mu_{i-1}))^2 f(\mu_i)}{(F(\mu_i) - F(\mu_{i-1}))^2} = 0,$$

by simplifying, we have

$$2\mu_i H(\mu_{i+1}, \mu_i) - H(\mu_{i+1}, \mu_i)^2 = 2\mu_i H(\mu_i, \mu_{i-1}) - H(\mu_i, \mu_{i-1})^2.$$

Since $H(\mu_{i+1}, \mu_i) > H(\mu_i, \mu_{i-1})$, it holds that

$$2\mu_i = H(\mu_{i+1}, \mu_i) + H(\mu_i, \mu_{i-1}).$$

$\square$

**Theorem 4.6.** *The coalition structure obtained from SPREAD is in-coalition core stable and individually stable.*

*Proof.* $\square$

**Theorem 4.4.** *There exists $n_0 = n_0(S, \alpha)$, such that the number of coalitions $n$ in the optimal coalition structure is not larger than $n_0$.*

*Proof.* We first show an upper bound of $E(\mathcal{C}^*_{n+1})$. Since $\mathcal{C}^*_{n+1}$ is the optimal coalition structure with minimum total error, we can bound this error by any other coalition structures. Hence, we focus on a class of truncated uniform coalition structure $\mathcal{C}^M_{n+1}$ where the federated model is chosen as the midpoint of an interval. Specifically, we choose $\mu_i = -R + \frac{2(i-1)R}{n-1}$ for all $i \in [n]$. It holds that

$$E[\mathcal{C}^M_{n+1}] = (n+1)S + V + \sum_{i=0}^{n} \int_{\mu_i}^{\mu_{i+1}} \left(x - \frac{\mu_i + \mu_{i+1}}{2}\right)^2 f(x)dx$$

$$\leq (n+1)S + V + \frac{R^2}{(n-1)^2} + \left(\int_{-\infty}^{-R} + \int_{R}^{\infty}\right) x^2 f(x)dx$$

$$\leq (n+1)S + V + \frac{R^2}{(n-1)^2} + \int_{|x|>R} Cx^2 e^{-\alpha|x|}dx$$

$$\leq (n+1)S + V + \frac{R^2}{(n-1)^2} + 2\exp^{-\alpha R}\left(\frac{R^2}{\alpha} + \frac{2R}{\alpha^2} + \frac{2}{\alpha^3}\right).$$

By choosing $R = (n-1)\sqrt{\frac{S}{2}}$, we have that

$$E[\mathcal{C}^M_{n+1}] \leq (n+1)S + V + \frac{S}{2} + 2\exp^{-\alpha(n-1)\sqrt{\frac{S}{2}}}\left(\frac{S(n-1)^2}{2\alpha} + \frac{\sqrt{2S}(n-1)}{\alpha^2} + \frac{2}{\alpha^3}\right).$$

Hence,

$$E(\mathcal{C}^*_{n+2}) - E(\mathcal{C}^*_{n+1}) \geq S - \frac{S}{2} - 2\exp^{-\alpha(n-1)}\left(\frac{(n-1)^2}{\alpha} + \frac{2(n-1)}{\alpha^2} + \frac{2}{\alpha^3}\right).$$

Hence, there exists $n_0$ such that when $n \geq n_0$, $E(\mathcal{C}^*_{n+2}) > E(\mathcal{C}^*_{n+1})$, so the number of coalitions in the optimal coalition structure is upper bounded by $n_0$. $\square$

**Lemma 4.5.** *For agent $a \in \mathbb{R}$, the rightmost agent $b(a)$ in her left-favorite coalition satisfies*

$$\frac{S}{F(b(a)) - F(a)} = 2(H(a, b(a)) - a)(b(a) - H(a, b(a))). \tag{10}$$

*Similarly, for agent $b \in \mathbb{R}$, the leftmost agent $a(b)$ in her right-favorite coalition satisfies*

$$\frac{S}{F(b) - F(a(b))} = 2(H(a(b), b) - a(b))(b - H(a(b), b)). \tag{11}$$

*Proof.* Without loss of generality, we assume that $V = 0$. For an agent $\mu$, the MSE in coalition $[a, b]$ is

$$l_\mu([a, b]) = \frac{S}{F(b) - F(a)} + (\mu - H(a, b))^2.$$

From the perspective of agent $a$, we define $L_a(b) = l_a([a, b])$ as the loss of agent $a$ when $a$ is the left end point of the coalition. The derivative of $L_a(b)$ is

$$L'_a(b) = -\frac{f(b)}{F(b) - F(a)}\left[\frac{S}{F(b) - F(a)} - 2\left(\frac{G(b) - G(a)}{F(b) - F(a)} - a\right)\left(b - \frac{G(b) - G(a)}{F(b) - F(a)}\right)\right]$$

$$= -\frac{f(b)}{F(b) - F(a)}\left[\frac{S}{F(b) - F(a)} - 2(H(a, b) - a)(b - H(a, b))\right].$$

Define $J(a, b)$ as

$$J(a, b) = \frac{S}{F(b) - F(a)} - 2(H(a, b) - a)(b - H(a, b)).$$

When $b \to a$, $J(a, a)$ goes to $\infty$; when $b \to \infty$, $J(a, \infty)$ goes to $-\infty$, so there exists $b^*$ such that

$$\frac{S}{F(b^*) - F(a)} = 2(H(a, b^*) - a)(b^* - H(a, b^*)).$$

To check whether $b^*$ is optimal for $a$, we consider the sign of $L_a'(b^* + \delta)$ where $|\delta| < \Delta$ for some constant $\Delta$. Note that $-\frac{f(b)}{F(b)-F(a)} < 0$ for all $b > a$, then we only need to consider the sign of $J(a, b^* + \delta)$.

The partial derivative of $J(a, b)$ with respect to $b$ is

$$\frac{\partial J(a, b)}{\partial b} = -\frac{f(b)}{F(b) - F(a)} \left( \frac{S}{F(b) - F(a)} - 2(H(a, b) - a)(b - H(a, b)) \right)$$
$$- \frac{2f(b)}{F(b) - F(a)} \left[ (b - H(a, b))^2 + (H(a, b) - a)\frac{F(b) - F(a)}{f(b)} \right]$$

so $\frac{\partial J(a,b)}{\partial b}|_{b=b^*} < 0$, then there exists $\Delta > 0$, such that $J(a, b^* + \delta) < 0$ and $J(a, b^* - \delta) > 0$ for all $\delta \in (0, \Delta)$. Hence, we have $L_a'(b^* + \delta) > 0$ and $L_a'(b^* - \delta) < 0$, which implies that $b^*$ is a local minimum point of $L_a(b)$. So the first part of Lemma 4.5 holds.

Similarly, we denote $R_b(a) = l_b([a, b])$ as the loss of agent $b$ when $b$ is the right end point of the coalition. The first derivative of $R_b(a)$ is

$$R_b'(a) = \frac{f(a)}{F(b) - F(a)} \left[ \frac{S}{F(b) - F(a)} - 2(H(a, b) - a)(b - H(a, b)) \right].$$

Till now, we see that similar ideas can be used to complete the proof of the second part of Lemma 4.5.

$\square$

**Theorem 4.6.** *The coalition structure obtained from SPREAD is in-coalition core stable and individually stable.*

*Proof.* Since SPREAD guarantees the boundary-optimality of the boundary agents, we see that no one can deviate to another coalition since it increases the utility of one of the boundary agent. Hence, the individual stability holds.

Now, we only need to focus on the in-coalition core stability. For a coalition $[l, u]$ formed by SPREAD, it holds that

$$\frac{S}{F(u) - F(l)} - 2(H(l, u) - l)(u - H(l, u)) = 0.$$

We assume that $[c, d] \subset [l, u]$ wants to deviate and form a smaller coalition. Note that $[c, d]$ must lies in the same side compared with $H(l, u)$ since agent $H(l, u)$ certainly does not want to deviate. We say that $[c, d] \subset [l, H(l, u)]$. Since they have motivation to leave, we have

$$\frac{S}{F(d) - F(c)} + (\tau - H(c, d))^2 \le \frac{S}{F(u) - F(l)} + (\tau - H(l, u))^2,$$

then

$$2\tau(H(l, u) - H(c, d)) \le \frac{S}{K} - \frac{S}{F(u) - F(l)} + H^2(l, u) - H^2(c, d).$$

As a result, we have that

$$2l(H(\mu, b_\mu) - H(c, d)) \le \frac{S}{K} - \frac{S}{F(u) - F(l)} + H^2(l, u) - H^2(c, d),$$

so $l$ can join $[c, d]$ to attain smaller loss, then $l$ can have even smaller loss in $[l, d] \subset [l, u]$, which contracts to the condition that $l$ has minimum loss in $[l, u]$. $\square$

**Theorem 5.1.** *If $P \ne NP$, there do not exist polynomial algorithms to solve the optimal coalition structure problem when $d \ge 2$.*

*Proof.* Assume for contradiction there is a polynomial-time algorithm $\mathcal{A}$ which, given $c$ and $M$, computes

$$L(c) \;=\; \min_{0 \leq x \leq M} \big( c\,x + f(x) \big)$$

and returns both the minimizer $x^*$ and the value $L(c)$. We show how to compute $f(x_0)$ for any $x_0 \in \{0, \ldots, M\}$ in polynomial time using $\mathcal{A}$, contradicting the NP-hardness of $f$.

Define a perturbed objective

$$g_\delta(x) = c\,x + \begin{cases} f(x) & x \neq x_0, \\ f(x_0) - \delta & x = x_0, \end{cases}$$

where $\delta \in [0, \Gamma]$ for some polynomial bound $\Gamma$ on the differences $|(c\,x + f(x)) - (c\,y + f(y))|$. Since $f$ is non-increasing and $c\,x$ is strictly increasing, the function $h(x) = c\,x + f(x)$ is unimodal, and subtracting $\delta$ at $x_0$ makes $x_0$ the unique minimizer of $g_\delta$ precisely when $\delta$ exceeds the gap

$$\min_{x \neq x_0} \big[ h(x) - h(x_0) \big].$$

Perform a binary search on $\delta \in [0, \Gamma]$ to find the smallest $\delta^+$ for which $\arg\min_x g_{\delta^+}(x) = x_0$. Each query to $\mathcal{A}$ takes polynomial time, and the search uses $O(\log \Gamma)$ queries, hence runs in polynomial time.

At $\delta^+$, $\mathcal{A}$ returns

$$L(\delta^+) = g_{\delta^+}(x_0) = c\,x_0 + \big( f(x_0) - \delta^+ \big).$$

Rearranging yields

$$f(x_0) = L(\delta^+) - c\,x_0 + \delta^+,$$

so $f(x_0)$ is computed in polynomial time, a contradiction. □

