# OpenReview forum: "Coalitional Personalized Federated Learning: A Hedonic Game Perspective"
_ICLR.cc/2026/Conference — ICLR 2026 Conference Withdrawn Submission_

### Official Review · Reviewer_JPkY · 2025-10-29

**Soundness:** 3
**Presentation:** 3
**Contribution:** 2
**Rating:** 4
**Confidence:** 4

**Summary:**

This paper studies the game-theoretic framework for federated learning using the solution concept of coalition formation, with key nolvety of addressing heterogeneous priors over agents' parameters. The theoretical analysis manily builds on exisitng work of Donahue & Kleinberg (2021), and the authors proposed 2 algorithms (BISCAN and SPREAD) for finding optimal and stable coalition structures in both atomic and non-atomic settings. The work is all theoretical analysis without any experimental validaditon.

**Strengths:**

The key novelty lies in generalizing Donahue & Kleinberg’s hedonic FL model to heterogeneous priors, providing constructive coalition-formation algorithms (BISCAN, SPREAD, which are computational efficient) with formal stability and optimality analysis.

**Weaknesses:**

1. The paper is entirely theoretical, with no experimental section or numerical simulation to substantiate the coalition-formation framework or to visualize stability and optimality in practice. As there is no empirical evidence showing that these procedures actually converge to the predicted stable coalitions, thoguh the authors introduce algorithms (BISCAN, SPREAD) for coalition formation, as they mianly conceptually and analytically.

2. The main conceptual foundation of this work—casting federated learning as a hedonic coalition game and analyzing stability vs. optimality are motivated and modeled by Donahue & Kleinberg (2021a,b); thus, rhe present “heterogeneous prior” generalization is mathematically incremental. In particular, the extensions analysis of atomic vs. non-atomic and algorithms (BISCAN, SPREAD) are logical continuations for analytical derivations rather than genuine breakthroughs.

3. What is the message to the FL community in terms of applicable and practical value, especially for FL clients perspectives, what should they do exactly, based on your work.

**Questions:**

1. what is the key model differs from Donahue & Kleinberg (2021b), e.g., how specific unique and important does heterogeneity of priors give, in terms of nolvety?
2. why no experiments, even synthetic validation, to validate the theoretical findings
3. How does the coalition structures compare to traditional clustering algorithms (k-means, hierarchical clustering)
4. what is the key contribution to FL community in terms of applicable and practical value

---

### Official Review · Reviewer_5FGK · 2025-10-30

**Soundness:** 2
**Presentation:** 3
**Contribution:** 2
**Rating:** 4
**Confidence:** 4

**Summary:**

This paper introduces a coalitional personalised federated learning framework using the hedonic game theory. The authors provide the methods for finding stable coalition structures and reduce the optimization problems to known formulations.

**Strengths:**

1. Generally, this paper is easy to follow.

2. The authors focus on an important problem in collaborative learning, coalition problem.

**Weaknesses:**

1. This paper is purely theoretical with no empirical experiments to validate the theoretical findings or demostrate practical applicability.

2. How does the proposed method embody federated learning, especially personalized federated learning, given that the paper’s background is entirely within this context?

3. In the related work section, all the cited studies in related to personlised federated learning were published before 2022. As personalized federated learning has gained significant attention over the past three years, with many new methods emerging, it would be beneficial for the authors to provide a more comprehensive review of recent literature in this area.

4. Some important works in the field of collaiton formation should be included, for example:

[1] Xiaohu Wu & Han Yu. MarS-FL: Enabling competitors to collaborate in federated learning. IEEE Transactions on Big Data, doi:10.1109/TBDATA.2022.3186991, IEEE (2022).

[2] Mengmeng Chen, Xiaohu Wu, Xiaoli Tang, Tiantian He, Yew-Soon Ong, Qiqi Liu, Qicheng Lao & Han Yu, "Free-Rider and Conflict Aware Collaboration Formation for Cross-Silo Federated Learning," in Proceedings of the 38th Annual Conference on Neural Information Processing Systems (NeurIPS'24), pp. 54974-55004, 2024.

**Questions:**

1. Why focus only on mean estimation? How would your approach work for more complex tasks like neural network training? Some discussion or even preliminary results on this would really strengthen the paper.

2. Can you characterize when multiple stable coalition structures exist? What are the properties of different stable outcomes?

3. How do your stability concepts hold when agents can arrive/leave over time or when data distributions drift?

---

### Official Review · Reviewer_SFw3 · 2025-10-30

**Soundness:** 2
**Presentation:** 3
**Contribution:** 2
**Rating:** 2
**Confidence:** 3

**Summary:**

This paper introduces Coalitional Personalized Federated Learning, a novel framework that models personalized federated learning through the lens of hedonic coalition games among self-interested agents. Unlike conventional federated learning, which aggregates all clients into a single global model, CPFL allows clients to form coalitions that collaboratively train specialized models aligned with their shared data distributions. Each client’s coalition choice is determined by a utility function based on the expected mean squared error under Bayesian priors. The study focuses on two main objectives: optimality, aimed at minimizing the total error across all agents, and stability, ensuring no agent has an incentive to deviate , under two different configurations: (1) atomic agents, for which the authors propose BISCAN (Bidirectional-Scan) to construct coalition structures satisfying both in-coalition core and individual stability; and (2) non-atomic agents, where the SPREAD algorithm achieves comparable stability guarantees within a continuous-agent framework.

**Strengths:**

- The paper offers a novel perspective by combining personalized federated learning with hedonic game theory. This framing is interesting, providing a principled lens to analyze coalition formation and stability.
- Reducing coalition optimization to regularized MSSC and optimal quantization is elegant. These links allow the problem to be grounded in well-studied mathematical frameworks.
- The paper demonstrates solid mathematical rigor, and the results appear to be correctly derived.

**Weaknesses:**

- **Here follow is my main concern**: Both BISCAN and SPREAD require as input each agent’s prior mean (expected optimal parameter under its local distribution). The framework thus assumes access to the very quantities it aims to estimate, a circular dependency that makes impossible to implement the currently stated method in the paper.

- The paper contains no experiments, no synthetic toy examples, no simulations, and no comparisons to baseline personalized FL methods. Without empirical evidence, it is impossible to assess whether the algorithms (or practical heuristics derived from them) yield useful coalitions.

- The related work section omits discussion of standard personalized federated learning approaches as well as prior clustering-based PFL methods. These methods directly address personalization and client heterogeneity in practical federated settings. The paper should clearly articulate what conceptual or empirical advantage CPFL provides over these existing approaches. For example, does the proposed framework improve personalization quality, fairness, or convergence efficiency? Without such a comparison or justification, it remains unclear why the hedonic-game formulation is needed or beneficial relative to well-established PFL baselines.

**Questions:**

- From my understanding, it's not possible to implement the exact algorithm as it depends on quantities that are intractable. Can the authors propose implementable versions of BISCAN and SPREAD? If so, can you propose an analysis of this implementable version?
- Could you compare how these two methods (or heuristic methods derived from them) perform against classical personalised federated learning methods on at least two different datasets?

---

### Official Review · Reviewer_H74F · 2025-10-31

**Soundness:** 2
**Presentation:** 2
**Contribution:** 2
**Rating:** 2
**Confidence:** 4

**Summary:**

This paper proposes a hedonic game view to coalition formation for personalized federated learning. It provides algorithms for clustering agents in both atomic and non-atomic (continuous) agent settings. In each case, it presents some theoretical results on the stability of the formed coalitions and their optimality (in terms of social optimality).

**Strengths:**

The idea of using game-theoretical methods to inform cluster/coalition formation in federated learning is a promising direction of work.

**Weaknesses:**

a. My first concern is around novelty and contribution. The paper states: “most existing papers mentioned in (Tan et al., 2022) are experimental researches without much theoretical analysis”. This seems inconsistent with a solid line of literature on theoretical guarantees of performance/accuracy/clustering algorithm convergence/communication overhead/etc., some examples of which are included below.

- The idea of providing formal performance guarantees for personalized learning is not new. There is a solid line of work establishing such performance guarantees. For instance, Werner et al. 2023 “Provably Personalized and Robust Federated Learning”, which proposes “simple iterative algorithms which identify clusters of similar clients and train a personalized model-per-cluster”, with convergence and performance guarantees on them. Similarly, Even et al. NeurIPS 2023 investigate “[…] Sample Optimality in Personalized Collaborative and Federated Learning”, considering agents with different local distributions and providing “matching lower and upper bounds on the number of samples required from all agents to approximately minimize the generalization error of a fixed agent”. There are many such examples with theoretical analysis and guarantees; I've only listed two samples that came immediately to mind for me, but I am confident there is plenty more.
- On a similar note, there seems to be a close similarity (if not identity) between clustering-based federated learning and coalition-based FL. There are a good number of papers with theoretical guarantees on personalized clustering-based FL. For instance, this paper (Deng, Yuyang, Mohammad Mahdi Kamani, and Mehrdad Mahdavi. "Adaptive personalized federated learning." arXiv preprint arXiv:2003.13461 (2020).), which has theoretical guarantees when clusters change adaptively, both on convergence of clustering assignments and their optimality. I understand that this one work does not claim that the perfect clusters have been formed (so they don't care about stability to my recollection), but as I argue in the last bullet, there should be more support as to why that is the right target for PFL. If the answer is: then agents would not stay in their assigned coalitions, my answer would be: there is also quite a bit of work about incentivizing participation in FL (both game-theoretical and otherwise. So there has to be a deeper reason, if any.


b. My next question is about how the error in (3) is defined, because that seems to be a central driver of clients’ goal is in forming coalitions and staying in them, and also in how social optimality is evaluated. First, it was unclear to me what S is, as S_k earlier is used to represent one random sample. Is it the trace of some matrix? Also, why is it assumed that agents know that/care about that? Second, it seems that is assumed that if x_C and x_k (prior expectation on the parameter) are far from each other (in l_2 norm), then there is higher “error”. This assumption needs support or justification. This definition of the error E is pretty much what seems to lead to the “Minimum Sum-of-Squares Clustering” (MSSC) problem formulation later, so providing support for it seems crucial. Finally, and perhaps most importantly here, how do the agents know or obtain x_C, which based on what I understand so far, is the *prior* expectation on the parameter by the cluster C?

c. Next, for the BISCAN algorithm, Theorem 3.5. is the core, but while I spent some time trying to understand its proof, I found it unclear and was not able to evaluate it for the following reasons. As one instance of lack of clarity, the proof makes mentions of “legal deviations”, and this term only comes up in this proof and not elsewhere in the paper, as far as I could find, and the term is not defined in the proof or elsewhere in the paper either. There is a mention of the “first phase” of the algorithm: does this mean the forward pass of the directional scan? Something else? The algorithm is not described with phases. Even outside of these issues with terminology which seem to be important to the proof, the claims in the proof of Theorem 3.5. require clarification and potential rewriting (at least I was unable to follow these claims). The proof of Lemma 3.4. has similar issues. Throughout its proof, a number of other lemmas and a corollary are proved, but it is not clear why each of them is presented in the middle of the proof of another lemma. Statements like “Actually, Corollary .9.1 completes the proof.” at the end of page 18 are an instance of added confusion for the reader, in the way the proof is presented. In summary, while the proofs might be correct, their exposition and lack of formality in the definitions/arguments prevented me from being able to check them. (I only tried to read the proof of Theorem 3.5 and Lemma 3.4., not the ones after, which might have similar issues.)

d. Lastly, and perhaps most crucially than the previous points, why I understand that this paper is aiming to provide theoretical support and algorithms, without support for its performance in at least some numerical experiment, it is hard to muster support for it. As I’ve noted above, there are other works with theoretical performance guarantees on different algorithms for personalized federated learning; most of them come with numerical experiments supporting that the algorithm is functional. Similarly, design choice of what the error function is, what knowledge agents have, etc., can vastly impact whether these algorithms are functional. Finally, if the paper is intended to be a purely theoretical exercise, then it should provide some generalizable theoretical insights or tools; again, I was not able to discern this.

**Questions:**

My 4 main questions are detailed in the weaknesses section. Please refer to those for the details of all questions. The ones below are only one sentence summaries (not the whole question).

a. What is the novelty of this work in light of the theoretical literature on personalized FL and clustering-based PFL?

b. How is the error function defined and well supported?

c. The proofs are unclear. Can they be clarified? (Though I acknowledge it would not be possible to redo all the proofs for a rebuttal, and this is more of a weakness that is not necessarily addressable by the "questions" section. I only leave it as a question in case the authors want to provide quick clarifications that may be able to fix the issues, though I suspect that is not feasible.)

d. Is there any support that the proposed algorithms can perform satisfactorily for PFL? If the paper is meant to be purely theoretical, what are the generalizable insights or methodologies or frameworks it is offering?

---

### Note · Authors · 2025-11-30

I have read and agree with the venue's withdrawal policy on behalf of myself and my co-authors.